# Optically addressable spin defects coupled to bound states in the continuum metasurfaces

Luca Sortino [1], Angus Gale[2], Lucca Kühner [1], Chi Li[3], Jonas Biechteler [1], Fedja J. Wendisch[1], Mehran Kianinia [2], Haoran Ren [3], Milos Toth [2,4], Stefan A. Maier [3,5], Igor Aharonovich [2,4] ✉ & Andreas Tittl [1] ✉

Van der Waals (vdW) materials, including hexagonal boron nitride (hBN), are layered crystalline solids with appealing properties for investigating light-matter interactions at the nanoscale. hBN has emerged as a versatile building block for nanophotonic structures, and the recent identification of native optically addressable spin defects has opened up exciting possibilities in quantum technologies. However, these defects exhibit relatively low quantum efficiencies and a broad emission spectrum, limiting potential applications. Optical metasurfaces present a novel approach to boost light emission efficiency, offering remarkable control over light-matter coupling at the sub-wavelength regime. Here, we propose and realise a monolithic scalable integration between intrinsic spin defects in hBN metasurfaces and high quality (Q) factor resonances, exceeding $10^2$, leveraging quasi-bound states in the continuum (qBICs). Coupling between defect ensembles and qBIC resonances delivers a 25-fold increase in photoluminescence intensity, accompanied by spectral narrowing to below 4 nm linewidth and increased narrowband spin-readout efficiency. Our findings demonstrate a new class of metasurfaces for spin-defect-based technologies and pave the way towards vdW-based nanophotonic devices with enhanced efficiency and sensitivity for quantum applications in imaging, sensing, and light emission.

Quantum systems incorporating vdW materials are highly desirable for practical device applications, supported by their capability to engineer novel heterostructures with hybrid and unique properties, enabling innovative approaches for integrated photonics[1,2]. Nanophotonic systems based on hBN have been successfully realized in various configurations[3], including photonic crystals[4], waveguides[5], and optical metasurfaces[6], further harnessing unique properties of hBN such as native visible quantum sources[7], and infrared phonon-polaritons[8]. Similarly, vdW materials have been employed as metasurfaces for non-linear optics[9], broad spectral tunability[6] as well as strong light–matter coupling[10], and hold great promise for technological scale-up via large-area production methods[11]. The identification of optically active spin defects in hBN[12,13] has paved the way for leveraging vdW materials as quantum optical systems[14]. This class of optically addressable spins[15] have emerged as a critical component of optical quantum technologies, functioning as qubits, sensors, and quantum repeaters[16]. Combining meta-optical elements, vdW materials, and optically active spin defects (Fig. 1a) represents an exciting frontier in the field of light–matter interaction[17], paving the way to unlock the intrinsic spin properties of the host materials for applications in quantum optics and quantum imaging.

[1]Chair in Hybrid Nanosystems, Nanoinstitute Munich, Faculty of Physics, Ludwig-Maximilians-Universität München, 80539 Munich, Germany. [2]School of Mathematical and Physical Sciences, University of Technology Sydney, Ultimo, NSW 2007, Australia. [3]School of Physics and Astronomy, Monash University, Wellington Rd, Clayton VIC 3800, Australia. [4]ARC Centre of Excellence for Transformative Meta-Optical Systems, University of Technology Sydney, Ultimo, NSW 2007, Australia. [5]The Blackett Laboratory, Department of Physics, Imperial College London, London SW7 2AZ, United Kingdom. ✉e-mail: Igor.Aharonovich@uts.edu.au; Andreas.Tittl@physik.uni-muenchen.de

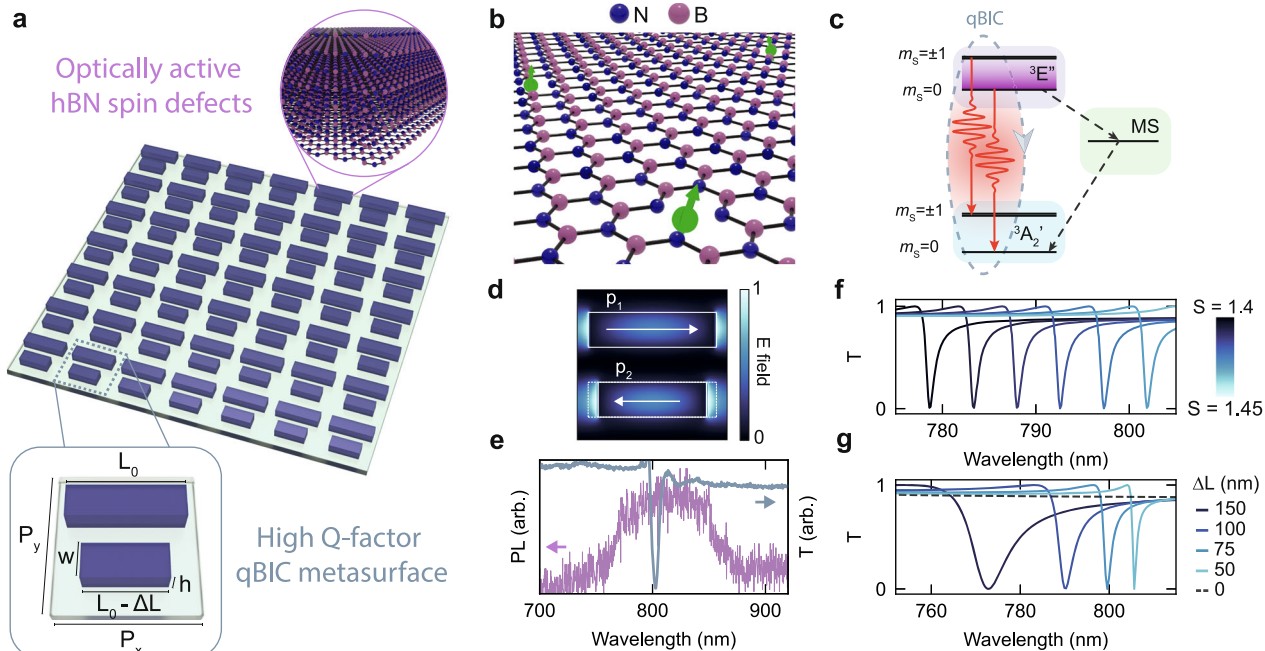

**Fig. 1 | Integration of intrinsic spin defects in hBN metasurfaces. a** Illustration of a symmetry broken qBIC optical metasurface fabricated from a single multilayer hBN crystal (top inset), on a glass substrate. The unit cell (bottom inset) is composed of two asymmetric rods. The asymmetry parameter is given by the difference $\Delta L$ in length between the two rods. **b** Illustration of the atomic structure of negatively charged boron vacancy defects ($V_B^-$, green arrows) depicted in a single layer of hBN. **c** Simplified energy level diagram of a $V_B^-$ defect with the excited state ($^3E''$), ground state ($^3A_2'$) and a metastable singlet state (MS). The qBIC acts as an optical cavity, resonantly enhancing the radiative transition of the coupled defect (gray arrow). **d** Electric dipole moments ($p_1$, $p_2$) and electric (E) field intensity for a symmetry broken unit cell, calculated at the qBIC resonance and excited with a plane wave linearly polarized parallel to the rod long axis. **e** Experimental PL spectrum of $V_B^-$ defects (in purple) and transmission spectrum (T) of a high-Q factor metasurface resonance (in gray). **f** Numerical FDTD simulations of the transmission of hBN metasurfaces for different scaling factors (S). Tuning of the scaling factor, and therefore the corresponding unit cell size, shifts the qBIC resonance wavelength over a broad spectral range. **g** Numerical FDTD simulations of the transmission of hBN metasurfaces for different values of the asymmetry parameter $\Delta L$. Increasing the asymmetry results in a blueshift of the qBIC resonance and a reduction of the Q factor. When $\Delta L = 0$ nm, the qBIC state transforms into a dark BIC state, and no resonance is present (dashed black line).

The negatively charged boron vacancy ($V_B^-$) is a promising spin defect in the vdW family, with an established crystallographic structure and a ground state spin triplet that can be initialized and readout at room temperature[12]. The $V_B^-$ can be implanted in any hBN flake via focused ion beams at length scales down to a few nanometers[18,19] and applied in quantum microscopy applications[20]. Figure 1b, c illustrate the atomic structure of the defect and a simplified energy level scheme of the $V_B^-$ spin states[21], respectively. The radiative transition is observed from the excited state ($^3E''$) to the ground state ($^3A_2'$), via a spin conserved transition. The ground state is a triplet, with a zero-field splitting $D_{GS}$ at ~3.5 GHz and spin coherence time up to 2 μs at room temperature[12,14]. However, the use of $V_B^-$ defects in practical quantum technologies is fundamentally limited by the low quantum efficiency, affected by intrinsic non-radiative processes and competing long-lived metastable states that limit the radiative transition rate. To overcome these drawbacks, enhancing the emission rate by coupling $V_B^-$ centers to optical resonators, such as dielectric antennas[22,23], nanobeam cavities[24], or plasmonic gap antennas[25–27] has been attempted. However, these approaches offered localized solutions lacking on-chip scaling prospects and yielding enhancements that are either broadband, with low control over the deterministic cavity-emitter coupling, or narrowband but limited by relatively low coupling efficiency. On the other end, optical metasurfaces offer a powerful platform and unprecedented opportunities for spectrally and spatially controlling light at sub-wavelength scales as well as large-area engineering and compatibility with industry-standard CMOS technologies[28,29]. Nevertheless, the combination of spin-photon functionalities into a unified metasurface structure with the versatile tuning of the emission enhancement has remained elusive.

Here, we demonstrate the integration and resonant coupling between optically active $V_B^-$ spin defects and high-Q factor resonances in hBN optical metasurfaces. Specifically, we leverage the physics of photonic bound states in the continuum (BICs)[30] to engineer high-Q factor cavities and realize precisely localized and strongly enhanced electromagnetic fields. Importantly, our approach is monolithic—that is, realized entirely from a single hBN crystal, simultaneously acting as the building block of the optical metasurface and as the defect's host material. Here, we demonstrate controlled fabrication and tuning of qBIC resonances in hBN metasurfaces and investigate the material-intrinsic enhanced light–matter interaction of the $V_B^-$ photoluminescence (PL) emission. We report more than one order of magnitude enhancement in the PL intensity, attributed to an increased local density of states associated with the resonant qBIC state. The coupling to the high-Q resonances results in a massive spectral narrowing of defect emission with a full width at half maximum (FWHM) below 4 nm. The qBIC-driven PL enhancement is realized by an in-plane field component of the qBIC resonance, revealing strongly enhanced coupling to the in-plane component of the dipole moment orientation of, still elusive, single $V_B^-$ defects. Finally, optically detected magnetic resonance (ODMR) measurements indicate that our hBN metasurfaces provide increased spin-readout efficiency combined with narrowband PL filtering. Our results demonstrate vdW materials as an exciting platform for novel fabrication strategies and enhancing the efficiency and sensitivity of room temperature quantum technologies. Moreover, our approach could be applied to more established spin defects in semiconductors, highlighting optical metasurfaces as exciting opportunities for light–matter interactions with spin-active defects.

## Results

Photonic BICs have established themselves as a new platform for engineering light–matter interactions at the nanoscale, with applications in diverse fields[31–34]. While "true" BICs are typically dark states, inaccessible for direct excitation from the far field, all-dielectric metasurfaces can leverage a breaking of the in-plane inversion symmetry of the meta-unit geometry to introduce a radiative loss channel that enables coupling to the incident light[31], yielding so-called quasi-bound state in the continuum (qBICs). These qBICs are observed spectrally as Fano-type resonances with high-Q factors approaching the infinite value of the true BIC state. We utilize a qBIC meta-unit composed of two asymmetric rod resonators, where the symmetry breaking is obtained by introducing a length difference $\Delta L = 50$ nm between the rods (Fig. 1a, inset). The unit cell geometrical parameters are base length $L_0 = 360$ nm, periodicities $P_y = P_x + 20$ nm $= 430$ nm, width $w = 100$ nm, and height $h = 160$ nm. Precisely tunable resonance wavelengths are obtained by applying a scaling factor S to all lateral dimensions of the unit cell, except for the height, which, in our case, is limited by the exfoliated crystal thickness. When illuminated with linearly polarized light oriented parallel to the longitudinal axis of the rods, the asymmetry gives rise to electric dipole moments with anti-parallel orientation in each rod[30] (Fig. 1d), introducing an effective dipole moment that can couple with the incident light. In analogy to established optical cavities, qBIC resonances with high-Q factors enable strongly enhanced light–matter interactions due to highly confined electromagnetic fields, enabling the efficient resonant coupling to the broad PL emission of $V_B^-$ defects (Fig. 1e). By controlling both the spectral shift of the qBIC resonance, induced by the geometrical variation of the unit cell (Fig. 1f), and the radiative Q factor, by tuning the asymmetry parameter (Fig. 1g), qBIC metasurfaces offer unprecedented opportunities to achieve critical coupling with luminescent emitters in sub-wavelength dimensions[10].

Figure 2a illustrates the fabrication process of the defective hBN metasurfaces. We first mechanically exfoliate thin layers (100–200 nm thickness) of single crystals of hBN on fused silica $SiO_2$ substrates. After selecting large crystals with suitable and homogeneous thickness, we design the matching qBIC resonances via full-wave numerical simulations and select the desired unit cell geometry. The sample is

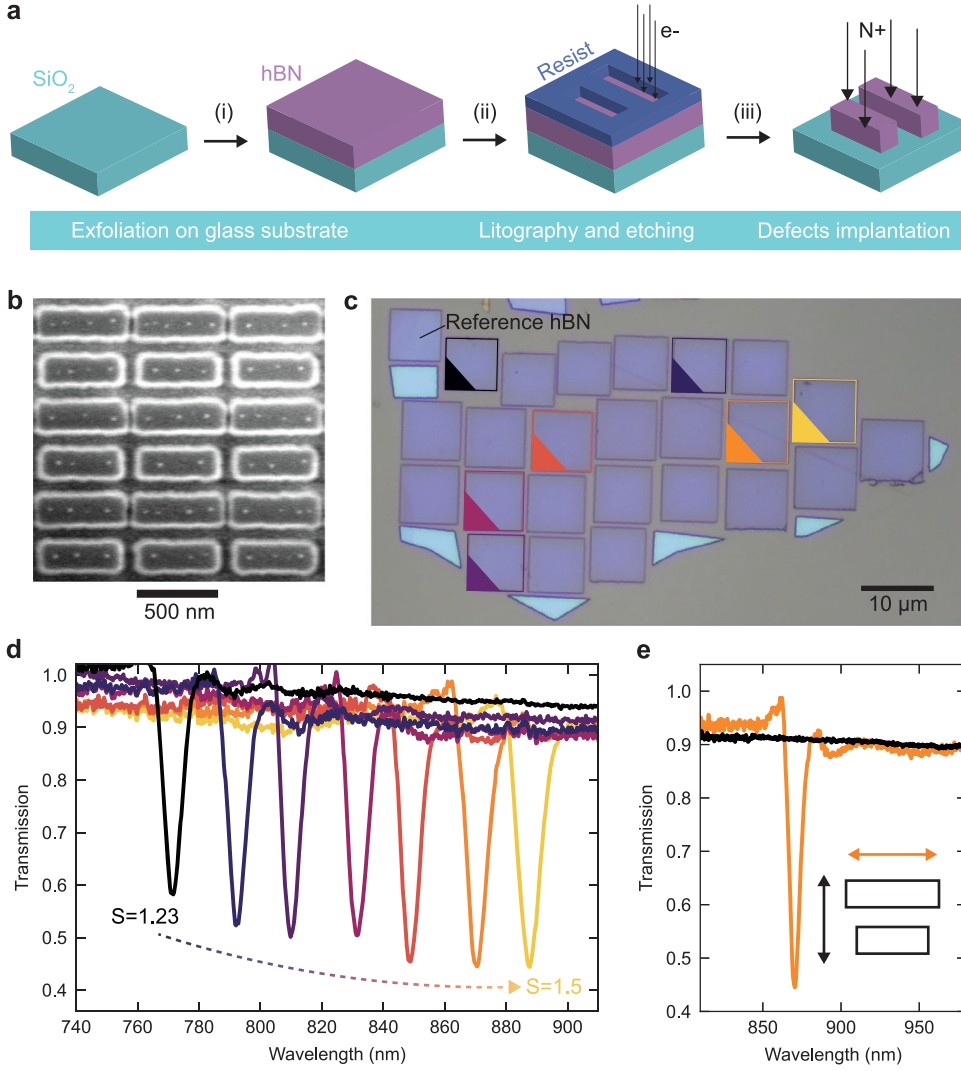

**Fig. 2 | Fabrication of defect implanted hBN metasurfaces. a** Fabrication process of the defect implanted hBN metasurfaces including hBN exfoliation from single crystals (i), electron beam lithography and reactive ion etching (ii), and nitrogen ion beam implantation of $V_B^-$ defects (iii) (see Methods). **b** Electron microscope image of spin-active hBN metasurface. **c** Bright field image of hBN metasurfaces with different scaling factors. Each square represents a single metasurface, and the brighter parts are leftover material of the pristine hBN layer. **d** Transmission spectra of the different hBN metasurfaces. The scaling factor varies from 1.23 to 1.5, exhibiting a redshift of the qBIC resonance with increasing unit cell size. The line colors correspond to the metasurfaces marked with the same color in the sample image in panel (**c**). **e** qBIC resonance spectrum for excitation under parallel (orange) and perpendicular (black) linear polarization.

structured via top-down nanofabrication methods, encompassing the two main steps of electron beam lithography and reactive ion etching (see Methods and ref. 6 for more information on the fabrication procedure). To generate the $V_B^-$ defects, the nanostructured hBN metasurfaces are irradiated homogeneously with a nitrogen ion beam (see Methods). Electron microscopy images confirm the accurate reproduction of the desired unit cell geometry of the hBN metasurface (Fig. 2b). Notably, we fabricated over 20 different metasurfaces with resonances spanning from around 750 to 900 nm, achieved by tuning the metasurface scaling factor from S = 1.23 to S = 1.5. As highlighted by the bright field microscopy image in Fig. 2c, this resonance tuning is obtained in a single exfoliated hBN crystal, with a height of 160 nm and lateral dimensions smaller than 100 μm × 50 μm. We analysed the white light transmission properties of hBN metasurfaces by means of confocal microscopy (see Methods). The transmission spectra of selected metasurfaces are shown in Fig. 2d, marked with the corresponding colors from Fig. 2c. As expected, we observe a redshift of the qBIC resonance to longer wavelengths for higher scaling factors, with Q factors ranging between 120 and 180 (Supplementary Note 1). Importantly, even though some structural defects of the pristine hBN layer are still present in the nanofabricated sample (see, e.g., the fold in the metasurface area marked in orange in Fig. 2c), we find that the transmission properties of the qBIC resonance are not significantly

affected when compared to resonances from arrays without structural defects (yellow metasurface area in Fig. 2c), as observed in Fig. 2d. To further confirm the qBIC nature of the high-Q resonance, we collected white light transmission spectra as a function of the excitation polarization angle. We clearly observed the sharp qBIC resonance when the excitation polarization was parallel to the longitudinal axis of the hBN rods, whereas the resonance was absent for perpendicular polarization since the longitudinal mode is not excited efficiently (Fig. 2e).

Figure 3a shows the white light transmission spectra of the complete set of fabricated hBN metasurfaces (see also Supplementary Note 2). The qBIC resonances exhibit a clear redshift with an increasing scaling factor, spanning from approximately 765 nm up to 890 nm and covering the entire spectral range of the $V_B^-$ PL emission. We probed the coupled $V_B^-$ PL emission in a back reflection geometry setup (see Methods), where the sample is excited by a continuous-wave laser excitation at λ = 532 nm, with linear polarization oriented parallel to the rods. As is shown in Fig. 3b, we observe that the $V_B^-$ PL signal collected for metasurfaces with different scaling factors exhibits a clear correlation with the qBIC resonance position, indicating strong resonant enhancement when both systems are spectrally coupled. To compare the effective number of defects excited under our experimental conditions, we normalized the reference spectrum by the filling factor of the unit cell, defined as the ratio between the unit cell area

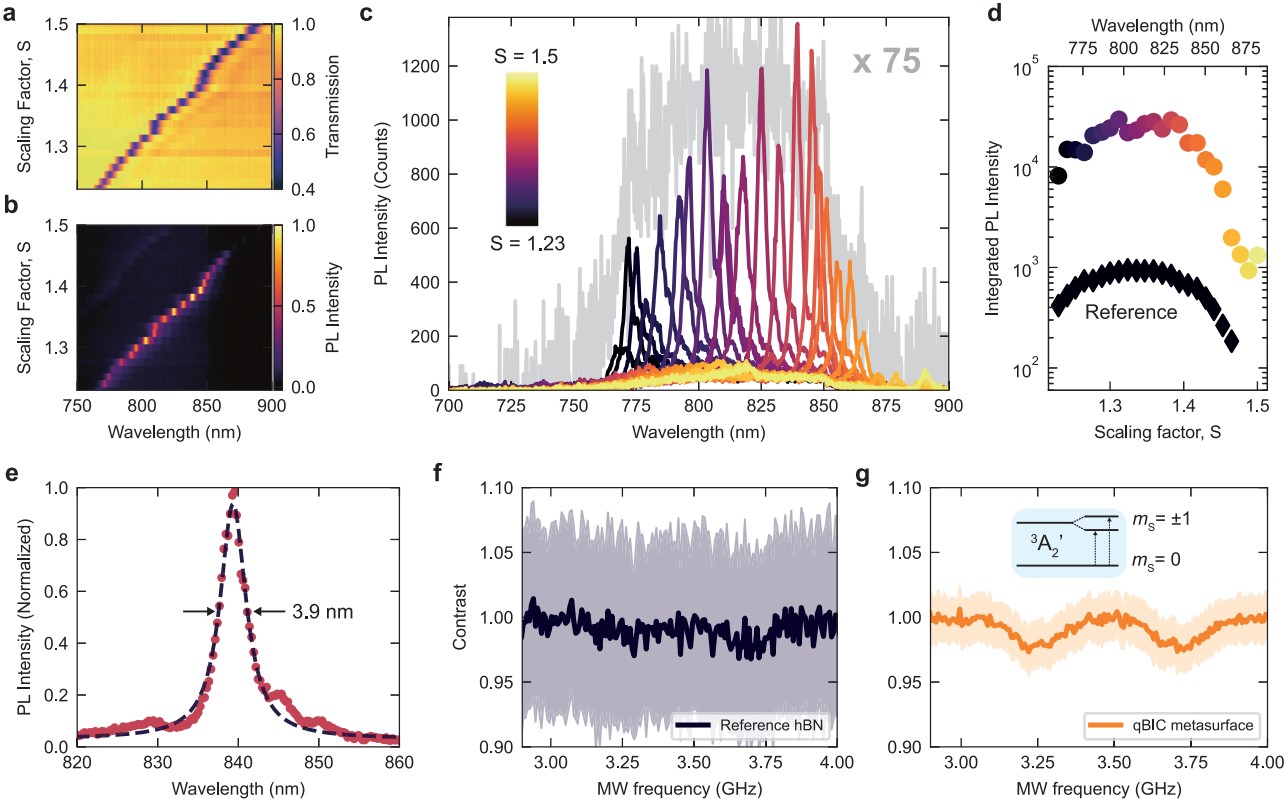

**Fig. 3 | PL enhancement of $V_B^-$ defects coupled to qBIC resonances and optical spin read-out. a** Experimental transmission spectra of the hBN qBIC metasurfaces as a function of the scaling factor S. **b** $V_B^-$ PL intensity when coupled to metasurfaces with varying scaling factors. The PL emission of the coupled defects is maximized for resonant coupling with the qBIC spectral position. **c** PL spectra of coupled defects for the full set of metasurfaces. In gray: reference $V_B^-$ PL spectra, normalized over the unit cell filling factor and multiplied by a factor of 75. The resonant enhancement of the qBIC metasurface shows excellent overlap with the $V_B^-$ PL emission spectrum. **d** Integrated PL intensity of the coupled system for different scaling factors. The PL is integrated over a 20 nm spectral window positioned around the PL maximum of the coupled defects. In black, the reference hBN integrated PL intensity values, normalized over the unit cell filling factor, obtained by

integrating over the same 20 nm spectral window. **e** PL emission from $V_B^-$ defects coupled to an hBN metasurface (in red) with a scaling factor of S = 1.39. The spectrum is fitted with a single Lorentzian peak (black dashed curve) showing a full width at half maximum (FWHM) below 4 nm. **f** Optically detected magnetic resonance (ODMR) contrast of the $V_B^-$ defects for the reference hBN crystal. The solid line represents the averaged values. **g** ODMR spectrum of $V_B^-$ defects coupled to a qBIC metasurface. The black dashed line is a fit to a two Lorentzian function, centered at 3.48 GHz. Inset: level scheme of the triplet ground state of the $V_B^-$ defect. Both spectra are integrated for 150 sweeps with an integration time of 1 ms at each microwave (MW) frequency. The PL is filtered with a 50 nm bandpass filter centered around 775 nm.

($A_O = (P_x * P_y)$) and the area defined by the hBN nanorods ($A_{BIC} = (L-\Delta L * w) + (L_O * x)$). Extracting the relevant geometrical parameters from SEM yields a filling factor of 0.4. The PL spectra acquired from the full set of fabricated hBN metasurfaces are plotted in Fig. 3c (colored lines), demonstrating excellent overlap with the $V_B^-$ PL emission spectrum (Fig. 3c, gray line), and over an order of magnitude enhancement in the $V_B^-$ PL emission. The variation of PL intensity between different qBIC modes can be attributed to the presence of resonant internal defect states and slight fabrication-induced structural inhomogeneities. Figure 3d shows the integrated PL intensity of the hBN metasurfaces for different scaling factors, adopting the color coding from Fig. 3d. The PL spectra are integrated over a 20 nm wide window covering the narrow PL emission of coupled defects and compared to the reference hBN PL emission in the same spectral range. We find that the integrated PL signals increase above one order of magnitude, yielding an average PL enhancement of 25.6 for the coupled defects (see Supplementary Note 3). Moreover, for all metasurfaces, the coupled emission exhibits an extremely narrow spectral band, favored by the high-Q factor resonances. As shown in Fig. 3e, we fit the PL emission of the coupled emission, yielding full width at half maximum (FWHM) values down to 4 nm. We further probe the ODMR signature of the $V_B^-$ spin defects, shown in Fig. 3f,g, under microwave (MW) excitation and employing narrowband filtering of the PL emission with a 50 nm spectral window. The noisy ODMR spectrum of the unstructured reference hBN area (Fig. 3f) precludes the detection of the triplet excited state splitting under bandpass filtering. In sharp contrast, when the defect is coupled to the qBIC metasurface (Fig. 3g), the ODMR exhibits an excellent signal-to-noise ratio (S/N = $P_{avg}/P_{std}$), defined as the ratio between the average value of the PL contrast ($P_{avg}$) and its standard deviation ($P_{std}$) for each MW frequency. For the coupled emission, we obtain the average value over the MW range of S/N = 28.5 compared to S/N = 6.9 for the reference sample (see also Supplementary Note 4 for a more detailed analysis of the S/N ratio), resulting in an overall 4.1 increase. The higher S/N ratio yields faster integration time with a small bandwidth, highly favorable for the efficient extraction of the ground state splitting and spectral selectivity in quantum sensing applications. The evidence of the ODMR response at approximately 3.5 GHz further confirms the optical activity of the electron spin resonance of the $V_B^-$ defects when coupled to the optical metasurface.

To elucidate the role of the resonant field enhancement in the qBIC metasurface, we further probed the PL emission intensity as a function of the excitation power. Figure 4a displays the PL spectra of an hBN metasurface with coupled spin defects for a scaling factor S = 1.31 and different excitation powers. By analyzing the integrated PL intensity over a small spectral window as a function of excitation power (Fig. 4b, inset), the different responses of coupled (Fig. 4b, yellow symbols) and uncoupled (Fig. 4b, blue symbols) hBN defects can be compared in detail. We fit the PL intensity power dependence with a linear function, where the slope of the linear fit represents the

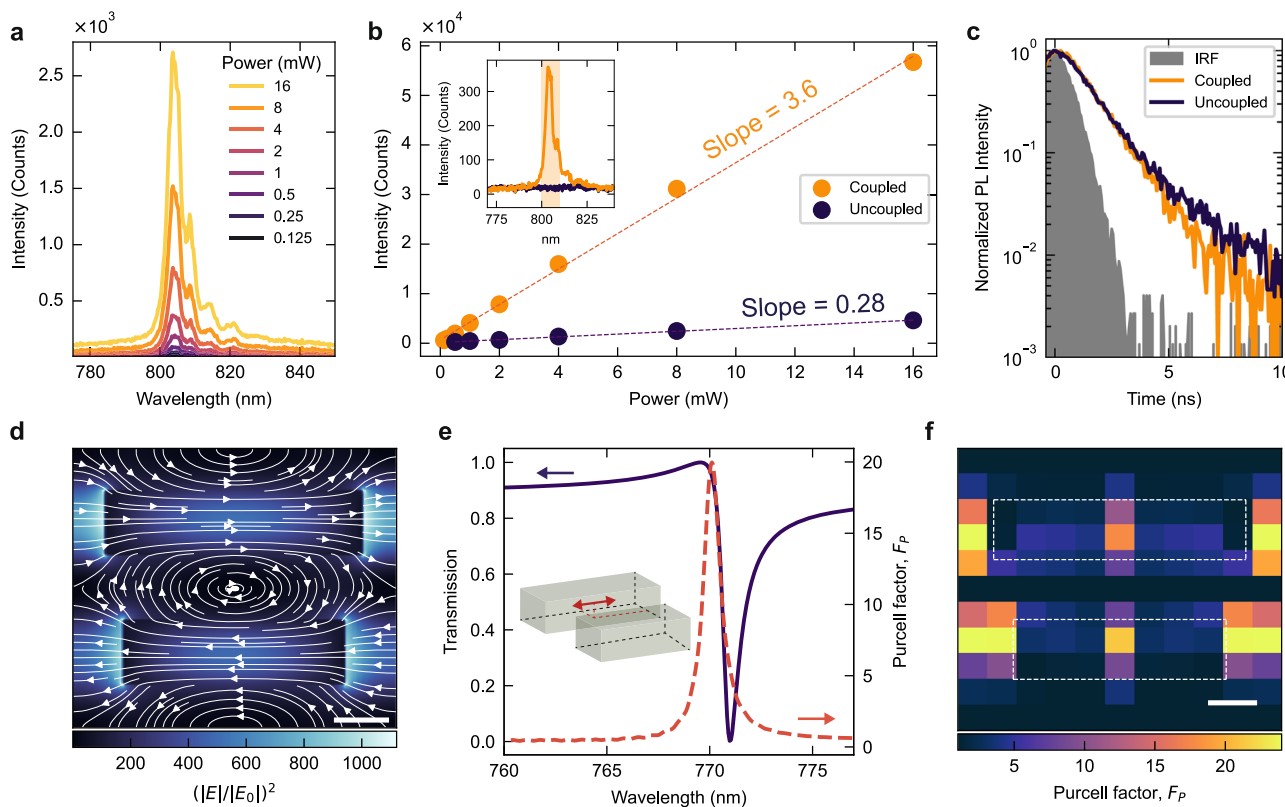

**Fig. 4 | qBIC-driven excitation of in-plane dipole emitters. a** PL spectra of $V_B^-$ defects coupled to a hBN metasurface with scaling factor S = 1.32, for increasing excitation power. **b** Integrated PL intensity as a function of the excitation power for coupled (yellow) and uncoupled (blue) defects. Inset: spectral window used for integration (shaded area) superimposed over the PL spectra of coupled (yellow) and uncoupled (blue) defects. **c** Time-resolved luminescence traces of coupled and uncoupled defect emission. **d** Numerical simulations of the electric field intensity for an hBN qBIC metasurface on a glass substrate. The monitor is placed at half of the height of the hBN thickness. The white arrows represent the vectorial field generated by the $E_x$ and $E_y$ field components. **e** Numerical simulation of the transmission (in blue) of a hBN metasurface with scaling factor S = 1.4 in vacuum and under parallel excitation polarization. In red, the Purcell factor ($F_P$) for an in-plane electric dipole. Inset: schematic of location and orientation of the dipole, placed at the center of the top nanorod of the hBN metasurface unit cell where the field is maximized. The $F_P$ value increases dramatically at the qBIC resonance, reaching values up to $F_P$ = 20. **f** Map of the Purcell factor calculated for a dipole oriented along the x-axis for different in-plane positions within the unit cell. The $F_P$ value is maximized in the central region of the two hBN nanorods, where the electric field enhancement is concentrated. Dashed white lines indicate the outlines of the nanorods.

change in emission intensity per unit increase of excitation power. We observe an increased excitation rate for coupled defects, with a slope of 3.6 compared to the reference hBN slope of 0.28. The slope ratio corresponds to an enhancement factor of ~12, indicating an increase in the overall quantum efficiency of the coupled defect. To confirm these observations, we collected time-resolved luminescence from coupled and uncoupled $V_B^-$ defects using a time-correlated single photon counting setup (Fig. 4c, see Methods for details). By convoluting a single exponential decay and the instrument response function (IRF), modeled by a single Gaussian peak, we extract lifetimes of $\tau_{on} = (1.287 \pm 0.23)$ ns for the coupled defects and $\tau_{off} = (1.552 \pm 0.29)$ ns for uncoupled defects in the reference hBN, which corresponds to a lifetime reduction factor of $\tau_{off}/\tau_{on} = 1.2$. Due to the long excitation pulse, the comparatively short defect lifetime, and the low quantum efficiency, the decay values extracted from luminescence experiments are heavily affected by the non-radiative processes, dominant over the inefficient radiative transition of the $V_B^-$ defects. For inhomogeneously broadened emitters coupled to a narrowband resonant cavity, the Purcell factor can be defined as[35] $F_P' = 1 + \gamma/\gamma_r [(\tau_{off}/\tau_{on}) - 1]$. Here, $\gamma/\gamma_r$ is the ratio of the population decay and spontaneous emission rate, which are intrinsic properties of each coupled defect, independent from cavity coupling. For an optically active defect, this ratio can be estimated from the quantum efficiency, the Debye–Waller factor, and the zero-phonon line branching ratio. However, owing to the low quantum efficiency of individual $V_B^-$ defects, knowledge of such parameters is not readily available. Nevertheless, funnelling PL emission with high-Q resonances contributes to increased indistinguishability for improved performance of room temperature operation of broadened quantum emitters[36], such as $V_B^-$.

We carried out numerical Finite Difference Time Domain (FDTD) simulations to elucidate the nature of the emitter-qBIC interaction. Figure 4d shows the electric field intensity together with the vectorial components (white arrows) of the in-plane electric field (see also Supplementary Note 5). The qBIC resonance offers strong field confinement, predominantly in the in-plane direction, affecting the properties of defects embedded in the hBN qBIC metasurfaces. Due to the orientation of the field, we focus on the maximally enhanced in-plane emitters. Figure 4e shows the calculated transmission spectra (in purple) of the qBIC hBN metasurface and the relative Purcell factor, $F_P$, (in red) for a single electric in-plane dipole, oriented parallel to the longitudinal axis of the nanorods and positioned at the center of the top nanorod (Fig. 4e, inset) where the field is maximized. The Purcell factor exhibits a significant increase close to the qBIC resonance, reaching values up to $F_P = 20$, indicating a substantial enhancement in the local density of states. Additionally, the enhancement is highly dependent on the dipole's position within the nanorod and orientation. The spatial map of the Purcell factor value for an in-plane dipole embedded in the hBN metasurface is shown in Fig. 4d. The maximum values are observed when the dipole is located at the center of the rods, corresponding to the region of the highest field enhancement. Following the orientation of the qBIC resonance field, we expect negligible coupling with out-of-plane dipoles, as shown in the z-component of the numerically simulated electric field in Supplementary Fig. 4c.

To establish a correlation between our numerical description and experimental results, we assume a uniform distribution of the defects, which would require the precise orientation of the embedded dipoles to calculate the coupling to the resonant in-plane electric field. However, experimental measurements of the dipole orientation of $V_B^-$ vacancies are still lacking, mainly due to the requirements of observable single defect emission. In our experiments, the correlation between the expected in-plane qBIC field enhancement, and the observed PL narrowing of the coupled defect emission, indicates a significant coupling to the in-plane component of the $V_B^-$ dipole orientation, consistent with what is observed in other hBN defect emitters[37]. To exclude possible effects of directional emission from the

metasurface, we carried out far-field calculations for resonant dipoles embedded in metasurfaces and in a reference hBN material, as shown in Supplementary Note 6. We do not observe relevant changes in the dipole radiation pattern, confirming the origin of the PL enhancement from the qBIC-driven light–matter coupling.

## Discussion

In conclusion, we have realized the monolithic integration of optically active spin defects with dielectric metasurfaces, leveraging the physics of photonic qBIC. Via the resonant interaction between optical modes and the $V_B^-$ defects, we observe enhanced PL emission leading to improved spin read-out efficiency. These results are enabled by the enhanced local density of states provided by the qBIC, resulting in over one order of magnitude of PL enhancement, together with spectral narrowing of the defect emission down to FWHM values below 4 nm, aided by the high-Q resonances. ODMR experiments shows the direct effect of the narrow and enhanced PL emission on the retrieval of the triplet spin state. The effect of PL filtering in our metasurface allows us to gather higher signal intensities and faster acquisition times compared to unfiltered detection. The use of spectral selectivity of our metasurface systems could find applications for hyperspectral imaging and sensing with luminescent defects. Moreover, via numerical simulations, we correlate the observed enhancement to a strong in-plane dipole moment of the $V_B^-$ defects, which has, so far, remained elusive to experimental verification. The convergence of metasurfaces and engineered light–matter interactions paves the way for new avenues in quantum optics based on coherent spin states. Our findings establish optical metasurfaces incorporating spin effects as a powerful platform for enhancing efficiency and sensitivity in optical quantum technologies, combining the advantages of mass production and CMOS compatibility. Our approach could be extended to other defects emitter or vdW-based systems, particularly hybrid ones, where magnetic domains or valley physics can be combined with quantum spin sensors and enhanced qBIC modes, for novel, on-chip nanophotonics applications.

## Methods

### Fabrication

hBN metasurfaces are fabricated from exfoliated bulk crystals following the method described in detail in ref. 6. The generation of $V_B^-$ defects is obtained by irradiating the hBN BIC metasurfaces in a Thermo Fisher Scientific Helios G4 Dual Beam microscope using a 30 kV nitrogen ion beam. The entire region was uniformly irradiated with a fluence of $1.0 \times 10^{14}$ ions/cm$^2$.

### Optical spectroscopy

White light transmission measurements are acquired in a home-built confocal transmission microscope setup. As an excitation source, a fiber-coupled supercontinuum white light laser (SuperK FIANIUM from NKT Photonics) is focused on the sample using a 10x objective (Olympus PLN, 0.25 NA) and collected using a 60x objective (Nikon MRH08630, 0.7 NA). Light is sent to a grating spectrometer and CCD camera with a multimode fiber (Princeton Instruments) for detection. PL measurements were taken using a confocal microscope in a back reflection geometry. A 532 nm continuous-wave (CW) laser excited the sample via a 10x objective (Nikon PLAN APO, 0.45 NA). A 532 nm long pass dichroic mirror and a 568 long pass filter were used to filter the reflected excitation laser before the spectrometer (Princeton Instruments). For transmission measurements, an additional laser path was aligned with CW 532 nm laser excitation focused onto the sample using a lens (Thorlabs AC254-100-A). The collection was taken with the reflection setup described above. For ODMR measurements, a 532 nm CW laser excited the sample via the same 10x objective. A copper wire was positioned within 10 μm of the fabricated structures and an RF signal generator created the microwave signal used for spin control. The collected photons were measured by an avalanche photodiode

(Excelitas Technologies). Time-resolved PL is acquired in a time-correlated single photon counting setup. A pulsed 512 nm laser with a 20 MHz repetition rate was utilized as the excitation source. A time correlator (PicoQuant PicoHarp 300) was used to synchronize the photon collection with an avalanche photodiode detector (Excelitas Technologies).

## Numerical simulations

Numerical FDTD simulations are carried out with commercial software (Ansys). The optical transmission of a $3 \times 3$ metasurface unit cell is obtained by illuminating the structure with linearly polarized light at normal incidence, and with periodic boundary conditions in the x- and y-directions. For simulations of the Purcell factor, we employed a broadband dipole modeled on the PL emission of the $V_B^-$ defects, with emission from 750 nm to 900 nm, oriented along the x-axis of the unit cell, parallel to the nanorods, and at a height of z = h/2. The Purcell factor is extracted as the power radiated by the dipole[38] collected with monitors positioned around the dipole, and calculated at different x,y-positions within the unit cell.

## Data availability

The data that support the findings of this study are available from https://doi.org/10.5281/zenodo.10450416.

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

## Acknowledgements

We are grateful to John Scott for developing the ion implantation technique, and Leonardo de Souza Menezes for fruitful discussions. We acknowledge Wirtshaus Atzinger for the conception of this study. Funded by the European Union (ERC, METANEXT, 101078018). Views and opinions expressed are however those of the author(s) only and do not

necessarily reflect those of the European Union or the European Research Council Executive Agency. Neither the European Union nor the granting authority can be held responsible for them. This work has received funding from the Deutsche Forschungsgemeinschaft (DFG, German Research Foundation) under Germany's Excellence Strategy (EXC 2089/1 – 390776260), Sachbeihilfe MA 4699/7-1 and the Emmy Noether program (TI 1063/1); the Bavarian program Solar Energies Go Hybrid (SolTech) and the Center for NanoScience (CeNS); the Australian Research Council (CE200100010, CE170100039, DE220101085, DP220102152, FT220100053) and the Office of Naval Research Global (N62909-22-1-2028). L.S. acknowledges funding support through a Humboldt Research Fellowship from the Alexander von Humboldt Foundation. H.R. and S.A.M. acknowledge the Australian Research Council (DP220102152). S.A.M. further acknowledges the Lee-Lucas Chair in Physics.

## Author contributions

A.T., I.A., L.S. and L.K. conceived the idea. L.S. and L.K. carried out the numerical simulations. L.K. and J.B. fabricated the hBN metasurfaces. F.J.W., L.K. and L.S. performed the optical characterization of the metasurfaces. A.G., C.L. and M.K. performed ion implantation and photoluminescence spectroscopy studies. L.S. wrote the manuscript with input from all the authors. I.A., M.T., H.R., A.T. and S.A.M. managed various aspects of the project.

## Funding

## Competing interests

The authors declare no competing interests.
