## [Peer Review File · Nature Communications]

I have reviewed the second response from the authors and am more confused than before. My previous question was not directly addressed. I am afraid that the whole S/N ratio enhancement argument is too arbitrary. The authors should be rigorous in data analysis and comparison. I would not recommend the publication in Nature Communications.

Here are the detailed comments:

1、 Here is my previous comment” The S/N ratio for the reference samples in Fig. 3f is supposed to be 6.9, but it looks less than that from the data quality. What is the signal and noise amplitude used here? The author also claimed more than one order of magnitude of enhancement with the spin defect coupled to the metasurface, which is 28.5 as claimed by the authors. Why 28.5 over 6.9 is more than one order of magnitude enhancement?” Please directly address why 28.5 over 6.9 is more than one order of magnitude enhancement.

2. Please check the figures below. Things just do not add up. Please be clear and address what is the defined S/N ratio in each case.

Fig. 1f is supposed to have an S/N ratio of 6.9, where does it come from? If that is true, what is the S/N ratio for Fig. 2a?

With the filtered configuration, why the data in Fig. 2b and 2c (with enhancement) is even worse than Fig. 2a (reference)? The resonance feature is completely gone in Fig. 2b and Fig. 2c, which are supposed to have an enhanced S/N ratio than Fig. 2a. Again, what is the defined S/N for each figure/trace?

Fig. 1. Figure in the previous round reply.

Fig. 2 Figures in the recent reply

Reviewers Comments:

Reviewer #1 (Comments for the Author):

I have reviewed the second response from the authors and am more confused than before. My previous question was not directly addressed. I am afraid that the whole S/N ratio enhancement argument is too arbitrary. The authors should be rigorous in data analysis and comparison. I would not recommend the publication in Nature Communications.

We thank the reviewer for their time in assessing our work. We believe that our reply below will resolve all remaining misunderstanding in our experiments and analysis.

Here are the detailed comments:

1、 Here is my previous comment” The S/N ratio for the reference samples in Fig. 3f is supposed to be 6.9, but it looks less than that from the data quality. What is the signal and noise amplitude used here? The author also claimed more than one order of magnitude of enhancement with the spin defect coupled to the metasurface, which is 28.5 as claimed by the authors. Why 28.5 over 6.9 is more than one order of magnitude enhancement?” Please directly address why 28.5 over 6.9 is more than one order of magnitude enhancement.

Our response:

We apologize for the confusion. We have addressed a similar point in the response to reviewer #1 in the previous round of reviews, which was also related to the S/N ratios, and which we thought would answer your questions as well. Here, we provide direct answers to all your questions:

“The S/N ratio for the reference samples in Fig. 3f is supposed to be 6.9, but it looks less than that from the data quality. What is the signal and noise amplitude used here?”

In our experiments, the noise amplitude is related to the fluctuations of the PL luminescence and detection electronics, calculated as the standard deviation of the PL intensity value, for each single microwave (MW) frequency, over the 150 integration scans of 100 ms each. The signal amplitude is then calculated as the average value of the PL intensity fluctuations. For Figure 3d, the average noise amplitude (PL standard deviation) over all MW frequencies is 0.142 and the signal (PL average) is 0.980, which yields a S/N of approximately 6.9 as already stated in the manuscript. This way of calculating the S/N is already clearly described in the manuscript on page 7, in the Methods and in Supplementary Note 4.

“The author also claimed more than one order of magnitude of enhancement with the spin defect coupled to the metasurface, which is 28.5 as claimed by the authors. Why 28.5 over 6.9 is more than one order of magnitude enhancement?”

Indeed, the enhancement factor for S/N is 4.1 as the reviewer states. This issue was already raised and replied to in the previous round of revisions and the current version of the manuscript does not contain any claims of one order of magnitude enhancement of the S/N. However, we stress that the *PL signal* shows a more than one order of magnitude enhancement between the reference and metasurface samples, which is mentioned many times in the manuscript and might have led to some confusion. Also, we want to highlight that even a 4-fold enhancement of the S/N is still useful for spin sensor applications, as it enables faster read-out speeds under small bandwidth detection.

Action taken:

In the previous round of revisions, we already softened the claim of a one order of magnitude enhancement of the S/N ratio. In addition, we now include the exact value of the S/N enhancement. These revised sentences now read:

“When the defect is coupled to the qBIC metasurface (Figure 3g), the ODMR exhibits an excellent signal-to-noise ratio ($S/N = P_{avg}/P_{std}$), defined as the ratio between the average value of the PL contrast (P_{avg}) and its standard deviation (P_{std}) for each MW frequency. For the coupled emission we obtain the average value over the MW range of $S/N = 28.5$ compared to $S/N = 6.9$ for the reference sample (see also Supplementary Note 4 for more detailed analysis on the S/N ratio), resulting in an overall 4.1 increase. The higher S/N ratio yields faster integration time with small bandwidth, highly favourable for the efficient extraction of the ground state splitting and spectral selectivity in quantum sensing applications.”

2. Please check the figures below. Things just do not add up. Please be clear and address what is the defined S/N ratio in each case.

Fig. 1f is supposed to have an S/N ratio of 6.9, where does it come from? If that is true, what is the S/N ratio for Fig. 2a?

Our response:

We believe that the confusion of the reviewer is related to Supplementary Figure S4. We provide in Supplementary Note 4 the analysis of the signal to noise ratio in our experiments, which were carefully taken with the same experimental configuration (integration time, excitation power, setup, etc.) to compare the effect of the PL enhancement in our hBN metasurfaces. We additionally present below an edited version of Supplementary Figure S4, which we believe is the source of misunderstanding.

Figure S4: (a-b) Unfiltered ODMR signal from the unstructured hBN reference sample (a) and a qBIC hBN metasurface (b). (c) Signal-to-Noise (S/N) ratio for unfiltered PL collection for the qBIC metasurface (orange) and unstructured sample (grey), averaged over 150 ms integration for each MW frequency. The dashed black line represents the average value. (d-e) Reproduction of Figure 3f-g from the main text of the manuscript: ODMR signal after a 775 ± 25 nm bandpass filter is placed in the collection path, from the unstructured hBN reference sample (d) and a qBIC hBN metasurface (e),

integrated over 150 ms for each MW frequency. (f) Signal-to-Noise ratio for Figures S4d-e. The dashed black line represents the average value.

In the case of the reference sample (Figure S4a), the higher number of luminescence centres and broad emission provides enough signal to observe an ODMR contrast. We compare both reference and metasurface unfiltered emission in Figure S4a-b, where we observe a clear ODMR signal for both samples. We then plot the S/N ratio for each MW frequency integrated over 150 ms in Figure S4c, and find an enhancement of the average S/N from average value of 21.3 to a value of 30.6 when going from the reference sample to the metasurface sample.

In Figure 2f of the manuscript, which is Figure 1f indicated by the reviewer and Figure S4d in the revised Supplementary Figure 4 above, the ODMR signal shown is now in the case of filtered VB-luminescence which provides too little signal to effectively observe the features of the triplet state splitting in the experimental configuration used. On the other hand, for the hBN metasurface the filtered signal provides high S/N ratio leading to the fast readout of the triplet splitting energy. This is then clearly observed in the S/N ratio plot in Figure S4f, where the 4-fold difference in the average values is observed.

Overall, our experiments show the direct effect of a narrow and enhanced PL emission from the coupled defects on the retrieval of an ODMR contrast. The effect of the PL filtering in our metasurface allows us to gather higher signal and faster acquisition times compared to unfiltered detection. Moreover, we note that the use of spectral selectivity of our metasurface systems could find application for hyperspectral imaging and sensing with luminescent defects.

With the filtered configuration, why the data in Fig. 2b and 2c (with enhancement) is even worse than Fig. 2a (reference)? The resonance feature is completely gone in Fig. 2b and Fig. 2c, which are supposed to have an enhanced S/N ratio than Fig. 2a. Again, what is the defined S/N for each figure/trace?

We believe this issue is due to a misunderstanding. Figure S4a displays the ODMR contrast, whereas Figures S4b-c (S4c,f in the revised version) display the S/N ratio for each MW frequency. Therefore, only Figure S4a is expected to show a resonance feature. For Figures S4b-c (S4c,f in the revised version), we then take the average value (dashed black line) to extract the S/N ratio for each sample and for filtered or unfiltered case. There are no features of the ODMR contrast in Figures S4b-c (S4c,f in the revised version) since these plots are not related to the relative PL intensity, but the relative ratio between average value and standard deviation in the experiments. The filtered and unfiltered case shows the overall increase in S/N ratio for metasurface coupled defects.

Action taken:

To better compare the ODMR signal, we provide an updated discussion in Supplementary Note 4, together with the revised Figure S4, as follows:

“The ODMR contrast for the unpatterned hBN reference sample is shown in Figure S4a, the data is integrated over 150 cycles with 1 ms integration time for each MW frequency. To compare with the patterned metasurface (Figure S4b), we define the Signal-to-Noise ratio ($S/N = P_{avg}/P_{std}$) as the ratio between the average value of the PL intensity (P_{avg} , solid line in Figure S4a,b) and the standard deviation of the PL signal at each MW frequency (P_{std}). We show the extracted value of this S/N ratio in Figure S4c. For unfiltered collection, we observe a small increase in the S/N ratio for the hBN metasurfaces from 21.3 to a value of 30.6. However, when inserting the 50 nm bandpass filter, we do not resolve a clear ODMR contrast anymore for the unstructured sample (Figure S4d). In contrast, the narrow emission from the metasurfaces funnels the light into the filter bandwidth and shows a clear ODMR

contrast (Figure S4e). In Figure S4f, we plot the S/N ratio for the filtered collection and with 150 ms integration time, showing a large increase compared to the unfiltered case. The S/N is enhanced from approximately 1.42 times in the unfiltered case, up to 4.13 in case of bandpass filter, as shown in Figures S4c,f, where we plot the S/N ratio for whole MW range investigated. The dashed black line represents the average S/N value.”

We have also added the following paragraph to the conclusion of the manuscript:

“ODMR experiments shows the direct effect of the narrow and enhanced PL emission on the retrieval of the triplet spin state. The effect of the PL filtering in our metasurface allows us to gather higher signal intensities and faster acquisition times compared to unfiltered detection. The use of spectral selectivity of our metasurface systems could find application for hyperspectral imaging and sensing with luminescent defects.”

REVIEWERS' COMMENTS

Reviewer #1 (Remarks to the Author):

After the analysis, the authors revised the manuscript and obtained an increased S/N ratio of 4 times instead of more than 10 times, and even that is only achieved for the filtered case. I believe that the results are not significant, and this work is better suited for a more specialized journal, not for Nature Communications.

Even the enhancement of 4 times is also tricky. It is for the filtered case by reducing the S/N of the unpatroned instead of increasing the S/N of the patterned BN metasurface. How useful it is for real application is debatable.

REVIEWERS' COMMENTS

Reviewer #2 (Remarks to the Author):

After the analysis, the authors revised the manuscript and obtained an increased S/N ratio of 4 times instead of more than 10 times, and even that is only achieved for the filtered case. I believe that the results are not significant, and this work is better suited for a more specialized journal, not for Nature Communications.

Even the enhancement of 4 times is also tricky. It is for the filtered case by reducing the S/N of the unpatterned instead of increasing the S/N of the patterned BN metasurface. How useful it is for real application is debatable.

Our response:

We appreciate the reviewer's feedback on the revised manuscript. However, we respectfully disagree with the assessment that the results are not relevant. A signal-to-noise improvement of four times is still noteworthy. Moreover, achieving this enhancement, even if for the filtered case, demonstrates a tangible advancement in our understanding of the behaviour of defects coupled to optical metasurface. Regarding the debate over the usefulness of the enhancement, we acknowledge that practical applications are a crucial consideration. However, the exploration of novel materials and their potential applications often involves incremental steps. Even if the enhancement is achieved by filtering, it can provide valuable insights into the underlying mechanisms at play, moreover, the narrowband enhancement in the photoluminescence emission of one order of magnitude can pave the way for further optimizations and potentially lead to practical applications in the future. Therefore, while we acknowledge the points raised, we believe that the revised manuscript still contributes meaningfully to the field.